# Homogeneous Incorporation of Gallium into Layered Double Hydroxide Lattice for Potential Radiodiagnostics: Proof-of-Concept

**DOI:** 10.3390/nano11010044

**Published:** 2020-12-26

**Authors:** Do-Gak Jeung, Tae-Hyun Kim, Jae-Min Oh

**Affiliations:** 1Department of Energy and Materials Engineering, Dongguk University-Seoul, Seoul 04620, Korea; jdk941101@naver.com; 2Department of Environmental Engineering, Seoul National University of Science and Technology, Seoul 01811, Korea

**Keywords:** layered double hydroxide, hydrothermal treatment, gallium, incorporation, diagnostics

## Abstract

Trivalent gallium ion was successfully incorporated into chemically well-defined MgAl-layered double hydroxide (LDH) frameworks through postsynthetic hydrothermal treatment. Quantitative analysis with inductively coupled plasma-mass spectroscopy exhibited that Ga^3+^ was first incorporated into LDH through partial dissolution-precipitation at the edge of LDH particle and homogeneously distributed throughout the particle by substitution of Ga^3+^ for Al^3+^ in LDH frame works. The powder X-ray diffraction patterns showed that the Ga^3+^ incorporation did not affect the crystal structure without evolution of unexpected impurities. The morphology and surface property of LDH evaluated by scanning electron microscopy and light scattering showed the preservation of physicochemical properties throughout 24 h of hydrothermal reaction. The distribution of incorporated Ga^3+^ was visualized with energy dispersive spectroscopy-assisted transmission electron microscopy, suggesting the homogeneous location of Ga^3+^ in an LDH particle. The X-ray absorption near-edge structure and extended X-ray absorption fine structure suggested that the Ga moiety was immobilized in LDH from 0.5 h and readily crystallized upon reaction time.

## 1. Introduction

For decades, layered double hydroxide (LDH) has attracted enormous interest as drug delivery carrier [1,2]. The LDH is composed of positively charged mixed metal hydroxide layers and charge compensating interlayer anions, with general chemical formula of M(II)_1-x_M(III)_x_(OH)_2_(A^n-^)_x/n_ mH_2_O, where M^2+^, M^3+^, and A^n-^ represented divalent, trivalent metal cations, and interlayer anion, respectively [3,4]. Due to the diverse metal composition and pH dependent solubility, LDH has been known to show high biocompatibility and biological inertness [5,6]. The unique properties of LDH such as anion exchange capacity, high cellular uptake, and controlled release of interlayer anion, allowed LDH to exhibit great potential as delivery carrier [7,8,9]. There have been extensive studies to encapsulate, stabilize, and deliver bioactive moieties such as anticancer drug [10], natural extract [11], antibiotics [2], nucleic acid [7], etc. utilizing LDH as a platform. In order to not only comprehend biological behavior of LDH carriers but also take advantage of diagnosis, many researchers have tried to incorporate tracing moieties into LDH for imaging [12]. Fluorescent dyes are one of the widely used labeling agents in laboratory level for in vitro and in vivo research due to high accessibility [13,14]. Recently, LDHs labeled with contrasting moieties, such as gadolinium (Gd) complex [15,16] and metallic nanoparticles [17] for magnetic resonance imaging (MRI) and X-ray computed tomography (CT), respectively, have been developed. However, those tracing moieties were bound to the external surface of LDH [17] or intercalated electrostatically between LDH layers [15,18], giving rise to stability issue of tracing agent under systemic circumstances. To address this stability issue, robust labeling such as direct incorporation of tracing moiety into the lattice of LDH has been arisen recently [19,20].

Among various labeling agent, radioisotope (RI) has emerged as a powerful tracer due to great sensitivity, outstanding tissue penetration, possible quantification, and excellent translation potential [21,22,23,24,25]. Currently, RI-based imaging such as single-photon emission computed tomography (SPECT) and positron emission tomography (PET) show high clinical accessibility. Taking into account the advantage of RI in imaging and delivery functionality of LDH, the incorporation of RI, e.g., ^57^Co or ^67^Ga into LDH frameworks became an option for diagnostic application [26,27,28].

However, direct incorporation of radionuclides into the LDH lattice for potential application in biological imaging is challenging. In order to take the maximum advantages of LDH in biological system, well-defined crystalline phase and uniform particle size are important factors [8,29,30]. Although LDH phase is easily formed within an hour through coprecipitating metal cations, thus prepared LDH used to have impurity and size problems. Fairly high content (~30%) of amorphous aluminum hydroxide, which is not detected by X-ray diffraction, was observed in the MgAl-LDH prepared by coprecipitation [31]. Unal et al. reported that preparation of MgGa-LDH without impurity—GaOOH or Ga_2_O_3_—requires high temperature (~200 °C) and long reaction time [32]. Furthermore, the coprecipitated LDH at short time produces small particles, which tends to form large aggregates [33,34]. In order to address the purity and particle size, LDH aimed for biological application is often prepared through hydrothermal reaction [35]. However, preparation of pure and particle size controlled LDH through hydrothermal reaction requires longer time, and thus, it is not a recommended method to incorporate radionuclide with short half-life (e.g., t_1/2_ of ^68^Ga and ^44^Sc are 68 min and 3.97 h, respectively) [36,37] into LDH. To overcome this time mismatch, Kim et al. reported direct incorporation of divalent ^57^Co, a single-photon emission computed tomography (SPECT)-sensitive RI, into the crystal lattice of LDH through hydrothermal substitution, which preserved morphology and crystallinity of LDH during short incorporation time [27,28]. Sufficient radiolabeling and tumor targeting efficacy were also addressed with the reported method, presenting the potential of LDH as radiodiagnosis.

This research might expand this concept to trivalent metal like gallium (Ga^3+^). As ^68^Ga is well known as PET imaging agent, the incorporation of gallium into LDH would suggest an alternative in contrasting agent. As a proof-of-concept experiment, this research proposed methodology to introduce nonradioactive Ga^3+^ into the LDH lattice preserving physicochemical properties. Through previous researches, incorporation methodology for certain metal species into layered metal hydroxide lattice, such as LDH [38] or brucite (Mg(OH)_2_) [39], was investigated. Divalent cation Co^2+^ was successfully incorporated into MgAl-LDH lattice through hydrothermal treatment-assisted isomorphous substitution for Mg^2+^ in the LDH lattice [38]. Furthermore, the Al^3+^ cations could be incorporated into Mg(OH)_2_ through partial dissolution-reprecipitation accompanying phase transformation from brucite to LDH [39]. Both cases showed successful introduction of target cations while preserving crystallinity and particle size. However, stabilized incorporation of Ga^3+^(aq) into LDH lattice at short time with preserved crystalline phase and particle size is another challenging project. Due to the strong acidity, the LDH is spontaneously dissolved upon encountering Ga^3+^ (aq), resulting in full dissolution-reprecipitation rather than incorporation of metal. In order to solve the problem, in this research, pretreated Ga^3+^ (aq) with NaOH to get Ga(OH)_4_^−^ species, which would react with LDH for incorporation, was utilized. Referring to the previous metal substitution method [27], Ga(OH)_4_^−^ and presynthesized LDH were reacted under hydrothermal condition. From this methodology, fast Ga^3+^ incorporation into LDH lattice were expected while preserving physicochemical properties of parent LDH such as morphology, surface charge, and crystallinity. To comprehend the detailed incorporation process of Ga^3+^ cation, time-dependent characterizations such as X-ray diffraction (XRD), field-emission scanning electron microscopy (FE-SEM), dynamic light scattering (DLS), electrophoretic light scattering, inductively coupled plasma-mass spectrometer (ICP-MS), energy dispersive spectroscopy-assisted transmission electron microscopy (TEM-EDS) mapping, and X-ray absorption spectroscopy (XAS) on designed time points (0.5, 1, 2, 3, 6, 12, and 24 h) were carried out. In the result and discussion section, this research will propose the incorporation mechanism first and demonstrate each aspect with corresponding characterization.

## 2. Materials and Methods

### 2.1. Materials

Magnesium nitrate hexahydrate (Mg(NO_3_)_2_·6H_2_O), aluminum nitrate nonahydrate (Al(NO_3_)_3_·9H_2_O), and gallium nitrate hydrate (Ga(NO_3_)_3_·xH_2_O) were obtained from Sigma-Aldrich Co. LLC. (St. Louis, MO, USA). Sodium hydroxide pellet (NaOH) and sodium hydrogencarbonate (NaHCO_3_) were acquired form Daejung Chemicals & Metals Co. Ltd. (Siheung-si, Gyeonggi-do, Korea). All reagents were utilized without further purification.

### 2.2. Synthesis of Parent MgAl-LDH

The parent MgAl-LDH with well-controlled particle size was synthesized by conventional coprecipitation and consecutive hydrothermal treatment. The mixed metal solution of Mg(NO_3_)_2_·6H_2_O (0.315 M) and Al(NO_3_)_3_·9H_2_O (0.105 M) was titrated with mixed alkaline solution of NaOH (1.2 M)/NaHCO_3_ (0.126 M) until pH 9.5. The obtained slurry was transferred to a Teflon-lined stainless-steel bomb and then reacted at 150 °C for 24 h. After then, precipitate was centrifuged, washed with deionized water, and lyophilized. The obtained parent MgAl-CO_3_-LDH powder was directly utilized for Ga^3+^ substitution though hydrothermal reaction.

### 2.3. Incorporation of Ga^3+^ into LDH

An aqueous suspension containing MgAl-LDH (5 mg/mL) and Ga(NO_3_)_3_·xH_2_O solution (0.05 M) was prepared separately. The Ga(NO_3_)_3_·xH_2_O solution was titrated by NaOH solution until pH 12 to produce Ga(OH)_4_^−^ as reactant. Then, LDH suspension was added to titrated Ga(NO_3_)_3_·xH_2_O solution and the mixture was hydrothermally treated in Teflon-lined stainless steel bomb at 150 °C. At each designed time points, 0.5, 1, 2, 3, 6, 12, and 24 h, the reaction was quenched and solid part was centrifuged, washed, and lyophilized. The Ga^3+^ incorporated LDH were represented as Ga@LDH-n, where n stands for the reaction time in hours.

### 2.4. Characterization

The XRD patterns of parent LDH and Ga@LDHs were obtained by X-ray diffractometer (SmartLab, Rigaku, Tokyo, Japan) with Cu K_α_ radiation (λ = 1.5406 Å) in the 2θ range of 5°–70° with scanning rate of 0.05°/s. Lattice parameters were calculated with UnitCell software (Tim Holland & Simon Redfern, 2006, Cambridge University, England) based on the obtained diffractograms. The crystallite sizes along the *c*-axis and *ab*-plane were calculated by Scherrer’s equation (t = 0.9λ/Bcosθ, t: crystallite size (Å), λ: X-ray wavelength, B: full-width at half-maximum of peak, θ: Bragg angle) [40] utilizing (003) and (110) peak, respectively. The contents of Ga^3+^ in Ga@LDHs were obtained by inductively coupled plasma-mass spectrometer (ICP-MS, NexION 2000, PerkinElmer, Middlesex, MA, USA). Hydrodynamic radii and zeta potential values of samples were examined by ELSZ-1000 (Otsuka, Kyoto, Japan) at 1 mg/mL concentration. The particle size and surface morphology of parent LDH, and Ga@LDHs were observed using field emission-scanning electron microscope (FE-SEM, JSM-7100F, JEOL, Tokyo, Japan). For SEM measurement, powdered samples were directly attached on carbon tape and then images were obtained with 15 kV acceleration voltage after Pt sputtering for 60 s. To obtain the average particle size and sample thickness, 50 particles were randomly selected from SEM images. In order to visualize the distribution of metal ions in Ga@LDHs, field-emission transmission electron microscopy (FE-TEM) and energy dispersive spectroscopy (EDS) mapping were performed (FE-TEM, Titan G2 ChemiSTEM Cs Probe, FEI, Hillsboro, OR, USA). To prepare TEM specimen, each powdered sample was dispersed into deionized water with ultrasonication for 10 min, and a drop of suspension was placed on 200-square mesh copper grid with carbon film. The TEM and EDS mapping images were acquired utilizing 200 kV accelerated electron beam. The local chemical environment around the incorporated Ga atoms was analyzed by X-ray absorption spectroscopy (XAS) at 8C NanoXAFS beamline in the Pohang Accelerator Laboratory (PAL), Pohang, Korea. All the powder-type samples were measured in transmission mode at Ga K-edge (10,375 eV). Normalized X-ray absorption near-edge structure (XANES) and extended X-ray absorption fine structure (EXAFS) spectra were obtained using Athena software.

## 3. Result and Discussion

### 3.1. Proposed Incorporation Mechanism of Ga^3+^ into LDH

Inspired by the previous substitution researches on LDH and brucite (Mg(OH)_2_) [38,39] and study on anion exchange from MgAl-CO_3_ to MgAl-NO_3_ under acidic condition [41], the Ga^3+^ incorporation mechanism were hypothesized that homogeneous incorporation of Ga^3+^ into the lattice of LDH would occur through following processes: (i) partial dissolution of LDH at the periphery, (ii) formation of amorphous Ga(OH)_3_ at the edge of LDH, (iii) migration of Ga^3+^ from the edge of LDH to the center by replacing Al^3+^, (iv) stabilization of LDH framework with balanced M(II)/M(III) ratio (Scheme 1).

First of all, Ga(OH)_4_^−^ anions approach the edge of LDH. As the solubility product (K_sp_) of Ga(OH)_3_ (7.28 × 10^−36^) is much lower than that of Mg(OH)_2_ (1.8 × 10^−11^) but fairly comparable with that of Al(OH)_3_ (1.99 × 10^−33^), the Ga(OH)_4_^−^ ion could precipitate at the edge of LDH (Scheme 1 (a)) by partially dissolving Mg(OH)_2_ moiety (Scheme 1 (b)). In this process, equilibrium of Equation (1) lies to the right. The precipitation of Ga(OH)_3_ and dissolution of Mg(OH)_2_ would be soon stabilized through dynamic equilibrium of Equation (2), developing thin layer of Ga(OH)_3_ at the edge of LDH particle.
Ga(OH)_4_^−^ (aq) + Mg(OH)_2_ (s) ⇌ Ga(OH)_3_ (s) + Mg(OH)_3_^−^ (aq)(1)
Ga(OH)_4_^−^ (aq) + Ga(OH)_3_ (s) ⇌ Ga(OH)_3_ (s) + Ga(OH)_4_^−^ (aq)(2)

Then the outermost Ga^3+^ is expected to migrate from the edge to the center of LDH particle through solid solution process [39,42] (Scheme 1 (c)), as we previously observed in Al^3+^ incorporation into brucite layers [39]. This process is mediated by substitution of Ga^3+^ for Al^3+^ to balance M(II)/M(III) ratio of which value 2.0~3.5 is reported to thermodynamically stable LDH structure [43]. The replacement of Al^3+^ by Ga^3+^ is rationalized by the in ionic radii: 76 pm for Ga^3+^, 67.5 pm for Al^3+^, and 86 pm for Mg^2+^. The comparable ionic radius of Ga^3+^ to Mg^2+^ would facilitate the stabilization of Ga^3+^ instead of Al^3+^ in the LDH lattice. The released Al^3+^ is considered to exist as solubilized ion as the basic reaction pH favors Al(OH)_4_^−^ species over Al(OH)_3_ precipitate.

### 3.2. Quantification of Metal Contents in Ga@LDHs

The chemical formulae of parent MgAl-LDH and Ga@LDHs (Table 1) supported the Ga^3+^ incorporation into LDH by replacing Al^3+^. The amount of Ga^3+^ compared with Al^3+^ in Ga@LDHs increased with respect to reaction time, suggesting the continuous incorporation of Ga^3^^+^. For quantitative analysis, time-dependent molar fraction of Ga^3+^ and Al^3+^ in Ga@LDHs was monitored based on the calculated chemical formulae (Figure 1A). The molar fraction of metal was defined by the ratio of certain metal over total metal content. The molar fraction of Ga^3+^ gradually increased from 0% to 2.6% along the reaction time (black line in Figure 1A), while the amount of Al^3+^ slightly increased from 20.3 to 21.8 but readily dropped to 19.8 (red dotted line in Figure 1A), showing continuous entering of Ga^3+^ into LDH by replacing Al^3+^ (Scheme 1 (c)). According to the increasing amount of Ga^3+^, the molar ratio of pre-existing metals (Mg^2+^ and Al^3+^) over supplied metal (Ga^3+^) dramatically decreased by 10.5 times (from 379.7 at 0.5 h to 36.5 at 24 h) (black line in Figure 1B), suggesting continuous supply of Ga^3+^ into LDH (Scheme 1 (b)). Interestingly, the metal ratio between divalent metal (Mg^2+^) and trivalent ones (Al^3+^ and Ga^3+^) in final Ga@LDH was equilibrated after abrupt decrease by ~11.5% at early time point (1 h) (red dotted line in Figure 1B). This may be attributed to the dissolution of Mg(OH)_2_ at the edge in early stage and thermodynamic stabilization through metal ratio balancing [43]. Under hydrothermal condition, the outermost low crystalline Ga(OH)_3_ could diffuse into the main framework of LDH by replacing Al^3+^ as often found in solid-solution process [44,45], resulting in the stabilization of M(II):M(III) ratio. From these quantitative analyses, we could find that hydrothermal treatment enabled incorporation of Ga^3+^ into LDH within 0.5 h and the homogeneous distribution followed along with lattice rearrangement occurred during 24 h.

### 3.3. Crystal Structure and Crystalline Impurities Analysis

It is generally known that Ga^3+^ tends to develop thermodynamically stable gallium oxide hydroxide (GaOOH) under hydroxide-rich condition [40]. In order to exclude the formation of unexpected Ga and Al-impurity during incorporation reaction, the PXRD patterns of samples at representative time points were recorded (Figure 2). The diffractogram showed that parent MgAl-LDH had typical hydrotalcite-like structure (JCPDS No. 14-0191) with sharp diffractions corresponding to (003) and (006) attributed to ordered 2-dimensional layer stacking [41]. In addition, the (110) and (113) diffraction from crystallographic *ab*-plane of LDH lattice were clearly observed at 60.4° and 61.7°, respectively.

The intensity of (003) reflection decreased along the reaction time until 3 h (Appendix A), which supported the partial dissolution of MgAl-LDH at early time point. After 3 h, the intensity of (003) diffraction gradually recovered upon homogeneous incorporation of Ga^3+^ in LDH structure. On the other hand, the reflections corresponding to *ab*-plane ((110) and (113)) apparently decreased. It is due to the evolution of partial dissolution of Al^3+^ during the substitution reaction [46]. Furthermore, XRD patterns of Ga@LDH shows only diffractions attributed by hydrotalcite without any possible impurities from Ga^3+^ and Al^3+^. It should be noted here that the lattice parameters of Ga@LDHs did not change significantly compared with parent MgAl-LDH (Table 1), suggesting that the global crystalline phase of LDH was not affected by the Ga^3+^ incorporation. To investigate the effect of reaction time on crystal structure in detail, the crystallite sizes along (003) and (110) direction were calculated by Scherrer’s equation. The crystallite sizes decreased around 6% and 41% for (003) and (110), respectively, during 2 h, suggesting the partial dissolution of early stage (Scheme 1 (b)). The crystallite size along *c*-axis recovered from 3 h and increased approximately 45% after 24 h due to the facilitated layer stacking under hydrothermal condition. On the other hand, the crystallite size of *ab*-plane only recovered 88% of parent LDH after 24 h, implying the possible development of defect in the lattice by Ga^3+^ incorporation. From the XRD analysis, it was found that post hydrothermal treatment for Ga^3+^ incorporation did not affect to the crystal structure of parent LDH and replacing Al^3+^ ions did not form any impurities and were removed by washing process.

### 3.4. Morphology and Surface Properties

Morphology is one of the most important parameters of LDH for various applications especially biomedical fields. The electron microscopic images of parent MgAl-LDH showed plate-like particles with uniform lateral size (Figure 3), which was the characteristic feature of hydrothermally prepared LDHs [33]. The characteristic plate-like morphology of LDH was well preserved throughout the 24 h of Ga^3+^ incorporation reaction (Figure 3c–h and Appendix A) without any irregular particles assuming Ga or Al impurities. The average lateral size of parent LDH determined was ~142 nm; taking into account the average value and standard deviation (Table 2), the lateral dimension of Ga@LDH did not significantly change regardless of reaction time. This result exhibited that the slight reduction in crystallite size along *ab*-plane was attributed not to the particle shrinkage but to the reduced intracrystalline arrangement, as discussed in Section 3.3. On the other hand, the thickness of Ga@LDH dramatically increased from 19 to 42 nm after 24 h of reaction time, along with crystallite size increase through *c*-axis (Table 1). From the TEM images, we also confirmed that parent MgAl-LDH and Ga@LDHs had platelets in the range of 100–200 nm without serious changes corresponded to SEM images.

The hydrodynamic radii of Ga@LDHs in aqueous suspensions were monitored by dynamic light scattering (Table 2). The hydrodynamic radius of parent MgAl-LDH determined was ~253 nm, suggesting that only two or three LDH particles were loosely agglomerated in aqueous state. The aggregation of LDH naturally occurs under water circumstances due to the strong edge-to-face interaction [47]. As hydrothermal treatment reduces unsaturated coordination site at the edge of particle [48], agglomeration among LDHs was fairly prevented. After Ga^3+^ incorporation, hydrodynamic radius of Ga@LDH showed similar value compared with parent LDH throughout 24 h of hydrothermal treatments. Under hydrothermal process, the smooth particle edge, i.e., reduction in unsaturated coordination, would maintain to prevent serious particle aggregation. It is known that the appropriate particle size of LDH facilitates cellular uptake by target cells [8], prolongs systemic circulation [30], and enables enhanced permeation and retention (EPR) [12,49,50]; therefore, stabilized hydrodynamic radius is an advantageous point of Ga@LDHs in biomedical application.

The positive surface charge is also one of the characteristic features of LDH, and thus, time-dependent zeta potential was monitored to comprehend surface chemistry of Ga@LDHs. The zeta potential of parent LDH was around +33.8 mV (Table 2), which well matched with the previous report [27]. The zeta potential slightly decreased at 2 h but readily recovered its positive value over +30 mV throughout 24 h (Table 2), suggesting the global surface characteristic of LDH maintained during Ga^3+^ incorporation reaction. It should be also noted that the zeta potential measurement let us exclude the formation of GaOOH impurity, which is thermodynamically stable under hydroxide-rich condition. It is reported that the GaOOH had negative zeta potential (−23 mV) in deionized water [51], which we could not find in our samples.

### 3.5. Distribution and Local Environments of Incorporated Ga^3+^ in Ga@LDH

The distribution of Ga^3+^ ion in the LDH after incorporation was visualized with scanning transmission electron microscopy (STEM)-energy dispersive spectroscopy (EDS) mapping (Figure 4 and Appendix A). To define the distribution of metal ions (Mg^2+^, Al^3+^, and Ga^3+^) in LDH and Ga@LDH, EDS mapping was carried out with high-angle annular dark-field (HAADF) image containing a few particles. As shown in Figure 4, green, yellow, and magenta colors indicated the location of Mg^2+^, Al^3+^, and Ga^3+^, respectively. Parent MgAl-LDH showed homogenous distribution of Mg^2+^ and Al^3+^ without development of magenta color. The incorporated Ga^3+^ was heterogeneously located in the edge of LDH particles at early time (Appendix A) as suggested in Scheme 1. After 24 h, (Figure 4B), the Ga^3+^ became homogeneously distributed throughout the LDH particle, supporting the time-dependent stabilization of LDH frameworks with balanced divalent (Mg^2+^) and trivalent (Al^3+^ and Ga^3+^) cations. This observation well matched with the quantification result summarized in Table 1.

To determine the local chemical environment around incorporated Ga^3+^, X-ray absorption near-edge structure (XANES) and extended X-ray absorption fine structure (EXAFS) were analyzed, as the techniques are sensitive and powerful method to investigate delicate changes in coordinate compound around a certain element. The XANES spectra of Ga@LDHs with main edge at ~10,370 eV (1s → 4p transition) [52] were fairly similar to that of MgGa-LDH—reference sample with stabilized Ga^3+^ in LDH structure—in terms of shape and sharpness, suggesting that the Ga^3+^ was well immobilized in LDH particles from 0.5 h. Slight difference in white line intensity indicated that the immobilized Ga^3+^ underwent chemical change inside the LDH particle. The white line intensity increment on Ga K-edge upon time was attributed to the strengthened Ga-O interaction through crystal rearrangement [53], in parallel with the finding in XRD analyses. At early stage of reaction, Ga^3+^ was considered to form low crystalline Ga(OH)_3_. Along the reaction time, Ga(OH)_6_ octahedron can be crystallized through homogeneous migration into LDH lattice (Scheme 1 (c)). After 24 h, Ga@LDH and reference sample, MgGa-LDH, showed almost identical white line intensity indicating that local structure around Ga^3+^ was stabilized. To precisely determine the main edge energy, first derivative of XANES spectrum was obtained (Figure 5B). It was found that the main edge energy of Ga@LDH shifted from 10,371.5 to 10,372.2 eV along the reaction time. This kind of shift was related to the increase in coordination number [54], which meant that Ga^3+^ migrated from defect site at the edge to crystalline site of center. In order to comprehend the local chemical environment around Ga^3+^ ions in Ga@LDH, R-space EXAFS spectra at Ga K-edge were analyzed (Figure 5C). It could clearly observe the first shell at 1.6 Å (nonphase-shift-corrected) and second peak at around 2.8 Å, which were attributed to the hexa-coordinate Ga-O and the neighboring metal ions such as Mg or Al [55,56]. It should be noted here that the FT amplitude of second shell is significantly lower than that of first shell (Figure 5C). According to the literatures, GaOOH [57] or Ga_2_O_3_ [58] showed both first and second shell peaks at similar position compared with MgGa-LDH. However, the FT amplitude of second shell in GaOOH or Ga_2_O_3_ was comparable with that of first shell due to the electron-rich nature of second shell Ga. The EXAFS spectra of current Ga@LDHs showed reduced peak at second shell, excluding the existence of GaOOH or Ga_2_O_3_. The XANES and EXAFS spectra revealed that incorporated Ga^3+^ ions were stabilized in the LDH frameworks with same local structure of Ga-containing LDH prepared by coprecipitation method. According to the suggested mechanism of Ga^3+^ incorporation (Scheme 1), Ga(OH)_3_ formed at early stage with low crystallinity transformed to better crystalline phase Ga(OH)_6_ upon reaction, reducing unsaturated coordination, which led to positive shift of edge energy. From the XAS results, it was found that around 0.5 h (shorter than half-life of ^68^Ga RI) of post hydrothermal reaction was fairly enough to immobilized Ga^3+^ moiety in LDH particle, whereas crystalline rearrangement took longer time.

## 4. Conclusions

The trivalent Ga^3+^ ions were successfully incorporated into MgAl-LDH lattice via postsynthetic hydrothermal treatment. The amount of incorporated Ga^3+^ gradually increased along the reaction time with decrement of Al^3+^ fraction. From quantification of metal ions in solid product, the incorporation of Ga^3+^ was closely related to the partial dissolution of Mg^2+^ moiety at first early stage and following replacement of Al^3+^ by Ga^3+^ in solid state. The absence of thermodynamically stable gallium phase, GaOOH, suggested that the reacted Ga^3+^ was well incorporated in LDH lattice. The morphology and surface properties of LDH were maintained throughout Ga^3+^ incorporation, suggesting that this method is advantageous labeling preserving physicochemical property of parent LDH. Visualized location of incorporated Ga^3+^ by EDS mapping indicated the homogeneously distribution starting from the edge of LDH particle. From XANES and EXAFS spectra, it suggests that Ga moiety was incorporated into LDH with low crystallinity but gradually crystallized upon hydrothermal treatment. Therefore, we could conclude that the postsynthetic hydrothermal treatment of LDH with trivalent metal ion was efficient way of metal incorporation at short time, which is especially required in RI tagging in LDH.

## Data Availability

The data presented in this study are available on request from the corresponding author.

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
