# Peer review of "Homogeneous Incorporation of Gallium into Layered Double Hydroxide Lattice for Potential Radiodiagnostics: Proof-of-Concept"

_nanomaterials, 2020, doi:10.3390/nano11010044_

Round 1

Reviewer 1 Report

The authors report on a formation of Mg-Al layered double hydroxide (LDH) with partial substitution of aluminium by gallium via a hydrothermal treatment of previously prepared Mg-Al LDH in a solution containing gallium nitrate at certain pH. Such a way of modification of the mixed-metal hydroxide layers of LDH looks very interesting and promising. Being thoroughly studied and verified, this way would open up new opportunities in production of layered hydroxides with the controllable cations content in the cases when direct synthesis of the LDHs with the desired combinations of divalent and trivalent cations is impossible.

However, in spite of the obvious importance of the topic, this particular paper has several serious drawbacks.

(1) The work is not really original. The same authors have already published a successful application of the post-synthesis hydrothermal treatment to formation of Mg-Al LDH from a mixture of magnesium hydroxide and aluminium hydroxide (Ref. [35]). The chemical reaction and the proposed mechanism are essentially similar to those reported in Ref. [35].

(2) The chosen objects of study are not convenient to demonstrate a possible advantage of the method of chemical modification of cation content of an LDH via hydrothermal treatment.  Both Mg-Al LDH and Mg-Ga LDH are very easy to produce by co-precipitation. A size difference between Al(III) and Ga(III) in octahedral coordination is small; therefore, a Mg-(Al,Ga) LDH with any rate of the controllable Al-to-Ga substitution LDH can be synthesized using the standard 1-step approach at room temperature.

(3) The assertion of the authors that the conventional synthesis of LDH takes too much time to be used for incorporation of radioisotopes into the hydroxide layers (lines 55-56) is not true. A standard synthesis of Mg-Al LDH via co-precipitation takes usually about one hour, which is comparable with the shortest period of the hydrothermal treatment applied by the authors. This makes the advantage of the method proposed in the manuscript rather doubtful.

(4) The chemical compositions of the obtained LDHs listed in Table 1 are misleading and do not correspond the general formula of an LDH indicated in line 30.

(5) No convincing evidence of incorporation of gallium in the sites of aluminium in the Mg-Al LDH has been provided. The results of the EDS and ICP-MS analyses cannot be considered as sufficient proof, because they indicate a distribution of the chemical elements in a sample, but not exactly in the LDH phase. In other words, the results of such analyses would be the same in the case of a homogeneous mixture of Mg-Al LDH + Ga(OH)3/GaOOH and in the case of a single-phase Mg-(Al,Ga) LDH. Moreover, the coordination of Ga(III) is the same in Mg-(Al,Ga) LDH, in Ga(OH)3, and in GaOOH, which makes the XAS technique not very indicative.

(6) The proposed mechanism of the Al-to-Ga substitution is uncomplete. For instance: if gallium substitutes aluminium, where does the “excess” aluminium go and in which chemical form?

English needs improvement. Some phrases are senseless or misleading. See, for instance, just in the 1st page:

«Trivalent gallium ion was successfully incorporated into particle size controlled MgAl- layered double hydroxide (LDH) lattice» (lines 12-13)

«Due to the versatile chemical composition and pH dependent solubility, LDH has been known to be highly biocompatible and biologically inert» (lines 32-33)

In my opinion, the manuscript is not acceptable for publication. It needs considerable revision and improvement both in motivation and novelty as well as in interpretation of the results and justification of the conclusions.

Author Response

Answers for the comments from Reviewer 1

The authors report on a formation of Mg-Al layered double hydroxide (LDH) with partial substitution of aluminium by gallium via a hydrothermal treatment of previously prepared Mg-Al LDH in a solution containing gallium nitrate at certain pH. Such a way of modification of the mixed-metal hydroxide layers of LDH looks very interesting and promising. Being thoroughly studied and verified, this way would open up new opportunities in production of layered hydroxides with the controllable cations content in the cases when direct synthesis of the LDHs with the desired combinations of divalent and trivalent cations is impossible.

However, in spite of the obvious importance of the topic, this particular paper has several serious drawbacks.

(1) The work is not really original. The same authors have already published a successful application of the post-synthesis hydrothermal treatment to formation of Mg-Al LDH from a mixture of magnesium hydroxide and aluminium hydroxide (Ref. [35]). The chemical reaction and the proposed mechanism are essentially similar to those reported in Ref. [35].

→ We appreciate kind comments of the reviewer 1. It is absolutely right that our previous researches – Co2+ incorporation into MgAl-LDH (Ref. [38]) and Al3+ substitution in brucite (Ref. [39]) – inspired the current research. This work has similarities and differences from the previous works as summarized below.

1) Compared with Ref. 38

Similarity: Substitution of a certain metal species for the lattice metal was carried out under hydrothermal condition. The phase was well-preserved due to the isomorphous substitution.

Difference: Previous work was done between divalent Co2+(aq) and MgAl-LDH, whereas current work focused on the reaction of trivalent Ga3+(aq) with MgAl-LDH. Both cations, Co2+ and Ga3+ are fairly different to each other in terms of acidity. The Ga3+(aq) readily dissolves LDH, while Co2+(aq) did not. Due to this difference, the substitution of trivalent metal into LDH lattice (especially basic one like MgAl-LDH) preserving the lattice structure is challenging.

2) Compared with Ref. 39

Similarity: Divalent metal species in brucite (Mg(OH)2) lattice was partially dissolved and trivalent metal was added to the lattice to produce LDH lattice. Partial dissolution-reprecipitation is expected for both cases.

Difference: Previous work engaged addition of higher valent metal (Al3+) and charge-compensating anion (CO32-) to drive phase transformation from brucite to LDH. While, the current research reports 1) the preservation of the phase and 2) the substitution of aqueous trivalent metal (Ga3+) for lattice trivalent metal (Al3+). Furthermore, the phase preservation was delicately controlled. In the Ref. 35, Al3+ solution was directly reacted with brucite under hydrothermal condition, where the brucite lattice did not dissolve significantly. However, when we first applied the method in the Ref. 34 for Ga3+ incorporation, MgAl-LDH was readily dissolved due to the strong acidity of Ga3+. In order to preserve the phase, we delicately find the condition, the utilization of less acidic Ga(OH)4- species for substitution.

We fully agree with the reviewer 1’s point that the present manuscript did not highlighted the differences compared with the previous literature. We revised the introduction part in order to explain the detailed originality of current research as follows from line 73 to 98.

(2) The chosen objects of study are not convenient to demonstrate a possible advantage of the method of chemical modification of cation content of an LDH via hydrothermal treatment.  Both Mg-Al LDH and Mg-Ga LDH are very easy to produce by co-precipitation. A size difference between Al(III) and Ga(III) in octahedral coordination is small; therefore, a Mg-(Al,Ga) LDH with any rate of the controllable Al-to-Ga substitution LDH can be synthesized using the standard 1-step approach at room temperature.

→ As pointed out by the reviewer 1, the phase of both MgAl-LDH and MgGa-LDH are easily obtained by conventional coprecipitation method. However, we would like to focus on two points, size and purity, to address why we did not consider the synthesis of MgGa-LDH directly by simple coprecipitation route.

Considering the biological application of LDH, the particle size between 100-300 nm the most advantageous in terms of cellular uptake and systemic targeting [8,29,35]. LDHs prepared by coprecipitation method used to have small lateral size (tens of nm) with rough surface, resulting in unexpected particle-particle aggregation and reduced biological performance. In order to take the maximum advantages in biological application, the LDH particles are usually prepared with long aging time or hydrothermal treatment. We are also aiming to utilize the current method to the Ga-68 incorporation into LDH and positron emission tomography application for the next research, it was required to produce LDH with 100-300 nm size.

Second point is the purity issue in coprecipitation method. It was recently reported that trivalent metal species produced unexpected impurity phase such as amorphous aluminum hydroxide (AOH) phase during coprecipitation. [31]. These impurities are not detected by XRD measurement but hinder further application of LDH. In order to deal with impurity problem, Unal et al. [32] investigated synthesis of MgGa-LDH by coprecipitation with various reaction conditions (time, temperature and with/without hydrothermal treatment). They claimed that MgGa-LDH without any impurities (GaOOH, Ga2O3) was obtained at high temperature (~200 ℃) and relatively long reaction time (at least 2 h).

For those reasons, we prepared MgAl-LDH with hydrothermal method, and tried to incorporate Ga3+. We agree with the reviewer that we should have mentioned why we excluded the direct coprecipitation of MgGa-LDH. We added relevant explanation in the introduction (line 54-72).

(3) The assertion of the authors that the conventional synthesis of LDH takes too much time to be used for incorporation of radioisotopes into the hydroxide layers (lines 55-56) is not true. A standard synthesis of Mg-Al LDH via co-precipitation takes usually about one hour, which is comparable with the shortest period of the hydrothermal treatment applied by the authors. This makes the advantage of the method proposed in the manuscript rather doubtful.

→ As pointed out by reviewer 1, it is absolutely right that LDH phase is readily obtained within an hour by coprecipitation method. We agree with the reviewer that the description in line 55-56 was ambiguous. In fact, tried to demonstrate that the importance of purity and particle size are important in biological application, and that the specific size and purity are hardly obtained with short time. As we replied in the comment (2), LDH prepared by coprecipitation method shows small particle size (related to unexpected aggregation) and amorphous purity [31,32]. To utilize LDH in biomedical applications, size and purity should be controlled uniformly, which takes longer time than short half-life time of radioactive Ga3+ (~68 min).

In order to clarify the point, we revised introduction part according to the response above (line 54-72).

(4) The chemical compositions of the obtained LDHs listed in Table 1 are misleading and do not correspond the general formula of an LDH indicated in line 30.

→ We appreciate kind advices of the reviewer 1. It is right that generally the chemical formula of LDH is written by two ways: 1) setting the mole number of hydroxyl as 2 or 2) utilizing integers for all the metal and hydroxyl. We agree with the reviewer that the general way is better to comprehend the metal ratios. However, we ask the reviewer 1 to understand that we intentionally set the mole number of Al as 1 to highlight the compositional change compared with Al3+ as we focused on the substitution of Ga3+ with Al3+. Furthermore, when we use the conventional notation, the number of digits for Ga becomes complicated due to the small molar fraction, hindering legibility. For example, chemical formula of Ga@LDH-0.5, Mg3.86AlGa0.01(CO3)0.5(OH)9.72 becomes Mg0.79Al0.21Ga0.0021(CO3)0.5(OH)2. One point we missed is that the interlayer anion in current chemical formula comes before hydroxyl. This is misleading and we corrected Table 1 in the revised manuscript.

(5) No convincing evidence of incorporation of gallium in the sites of aluminium in the Mg-Al LDH has been provided. The results of the EDS and ICP-MS analyses cannot be considered as sufficient proof, because they indicate a distribution of the chemical elements in a sample, but not exactly in the LDH phase. In other words, the results of such analyses would be the same in the case of a homogeneous mixture of Mg-Al LDH + Ga(OH)3/GaOOH and in the case of a single-phase Mg-(Al,Ga) LDH. Moreover, the coordination of Ga(III) is the same in Mg-(Al,Ga) LDH, in Ga(OH)3, and in GaOOH, which makes the XAS technique not very indicative.

→ As pointed out by reviewer, the quantification results by ICP-MS show only the increment of Ga3+ in Ga@LDH solid product. There are two major points we highlighted in the original manuscript to suggest the incorporation of Ga3+ in the lattice of LDH rather existence of Ga-phase as mixture with LDH. i) Potential impurities such as Ga(OH)3 or GaOOH, which are usually crystalline after hydrothermal treatment, were not observed in XRD patterns (Fig. 2). ii) Surface charge of GaOOH under aqueous suspension is negative. In spite of the homogeneous Ga distribution at the surface of LDH (EDS mapping result in Fig. 4), we could not observe significant zeta potential change from positive to negative direction in Ga@LDHs (Table 2). The results implied that there is low possibility of GaOOH formation on LDH surface.

In addition to the above points, we further analyzed extended X-ray absorption fine structure (EXAFS) to examine the local structure around Ga3+, as we agree with the reviewer 1 that XANES of (GaOOH + MgAl-LDH mixture) is not easily distinguished from that of Ga-incorporated LDH. As shown in the Fig. 0 (A) below, the R-space spectra of Ga@LDHs as well as MgGa-LDH showed a strong first shell peak at 1.6 Å (non-phase-shift-corrected) and a weak second shell peak at 2.8 Å. As the reviewer pointed out all the Ga-O species including MgGa-LDH, Ga2O3, and GaOOH has similar Ga-O (first shell) distance and Ga-M (second shell) distance. However, there is a clear difference in terms of the FT amplitude of second shell. As the GaOOH and Ga2O3 definitely have Ga-Ga environment in the second shell, the corresponding amplitude is very strong (Fig. 0 (B-C)). On the other hand, the second shell of MgGa-LDH and Ga@LDH is Ga-Mg, the FT amplitude becomes smaller than that of first shell. The clear reduction in the FT amplitude implied that there is less possibility of Ga occupying the second shell position. The result also supported the low possibility of GaOOH or Ga2O3 in Ga@LDH samples.

Figure 0. (A) Fourier-transform extended X-ray absorption fine structure (FT-EXAFS) spectra of Ga@LDH-0.5 (black line), Ga@LDH-1(red line), Ga@LDH-24(blue line) and MgGa-LDH (green dots). (B) R-space and k-space EXAFS spectra of Ga2O3 (reproduced from RSC Adv., 2017, 7, 52543-52554) and (C) R-space EXAFS spectra of GaOOH (reproduced from Geochimica et Cosmochimica Acta, 2002, 66, 4203-4222)

We added the R-space EXAFS spectra of Ga@LDHs as Fig. 5 (C) and added relevant discussion in line 309-320.

(6) The proposed mechanism of the Al-to-Ga substitution is uncomplete. For instance: if gallium substitutes aluminium, where does the “excess” aluminium go and in which chemical form?

→ We appreciate kind comments of the reviewer 1. As the reviewer pointed out, the proposed mechanism of Ga3+ incorporation into LDH frameworks is not the completed one. Rather it is a proposed hypothesis of Ga3+ incorporation. There should be more detailed points to be addressed; however, several aspects of the four-steps could be explained by the data we displayed. For example, we hypothesized that Ga3+-incorporation was mediated by both adsorption of Ga3+ on LDH’s surface along with partial dissolution of Mg(OH)2 and the Ga3+ substitution for Al. Addition of Ga3+ into LDH was demonstrated with the increasing Ga molar fraction in LDH (Fig. 1 (A) black solid line). Partial dissolution of Mg(OH)2 is both rationalized by the pH-dependent solubility of Mg and the reduced M(II)/M(III) molar ratio (Fig. 1 (B) red dotted line). The substitution of Ga3+ for Al3+ is denoted as the time-dependent increase in Ga content (Fig. 1 (A) black solid line) along with the reduction in Al3+ content (Fig. 1 (A) red dotted line). Although, there observed temporary increase in Al fraction, it was attributed to the relative increase by Mg-dissolution. In fact, the final Al3+ fraction decreased by 0.5%, suggesting the dissolution of Al. In order to clarify that Al3+ could be release through substitution, we slightly modified the scheme 1 by indicating the potential release out of Al3+.

As pointed out by the reviewer, the status of aluminum should be addressed. We think that most of the released Al3+ existed in the solution as ions. The reaction condition of current substitution is fairly basic (pH > 11), in which condition, Al(OH)4-(aq) ions dominantly exist compared with Al(OH)3. There is less chance for Al3+ to re-enter LDH framework as the M(II)/M(III) ratio in the LDH lattice is usually stabilized in 2.0-3.5 range. In order to clarify the point, we indicated that the released Al3+ could be solubilized as Al(OH)4- form in the revised manuscript (line 171-172).

English needs improvement. Some phrases are senseless or misleading. See, for instance, just in the 1st page:

«Trivalent gallium ion was successfully incorporated into particle size controlled MgAl- layered double hydroxide (LDH) lattice» (lines 12-13)

«Due to the versatile chemical composition and pH dependent solubility, LDH has been known to be highly biocompatible and biologically inert» (lines 32-33)

→ We appreciate kind comments of the reviewer 1. We thoroughly revised the manuscript including the points raised by the reviewer.

In my opinion, the manuscript is not acceptable for publication. It needs considerable revision and improvement both in motivation and novelty as well as in interpretation of the results and justification of the conclusions.

→ We appreciate the reviewer 1’s kind comments to improve current manuscript. According to the advices, we realized that there are points to be explained in detail and data to be re-interpreted. The indicated points were sincerely reflected in this letter and in the manuscript. We wish that the revised version meets the standard of Nanomaterials for the publication.

Reviewer 2 Report

The work elegantly and clearly demontrated the incorporation of Ga within LDH. The rpesented data support the conclusions. For these reason, the manuscript can be acceprted for publication, provided that the below listed minor comemnts are solved.
As a final note, a general improvement of the language is needed. In particular (see minor note 1 as example), the use of plural must be preferred to singular form everywhere in the manuscript. Also, avoid within all the manuscript the use of first person (for instance, line 78 “we will propose..”)

Minor notes:
1) line 39 use “dyes are … agents” instead of “dye is … agent”
2) Line 43 “labeled on” is not the correct term: “bound to”?
3) Line 75-76 “field-75 emission scanning electron microscopy (FE-SEM),” is cited while in the abstract trasmision EM is cited: In experimental both trasmission and scanning: please clarify and make consintent all the sections
4) Line 78: “from 0.5 h to 24 h” explain better;
5) Line 83 “Ga(NO3)3·xH2O” what about “x”? Please explain the meaning
6) Line 132 The partial dissolution of crystallites during LDH anion exchange reactions was demsotrated by Palin et al., 2019, Understanding the Ion Exchange Process in LDH Nanomaterials by Fast In Situ XRPD and PCA-Assisted Kinetic Analysis, Journal of Nanomaterials, https://doi.org/10.1155/2019/4612493: the work, where the kintic of the processi s also discussed, should be cited and commented here and/or in the discussion
7) Line 149 “insertion” instead of “stabilization”?
8) Line 154: in the caption of figure 1, please write the technique used for Ga quantification
9) Line 177: plase in the caption of table 1 plese recall the maning on sample names “Ga@LDH-x”. i.e the meaning of the number “x”; in the text recall how these samples were produced, maybe referring to the experimental section. Remove the first decimal digit from crystal sizes and explain that this is an estimation, known all the limitations of Sherrer approach, expecially when applied to layered materials. Please comment the differences with sizes from electron microscopy
10) Line 180 the title of section 3.3 is misleading. Maybe “Crystalliniy and purity analysis”?

Author Response

Answers for the comments from Reviewer 2

The work elegantly and clearly demontrated the incorporation of Ga within LDH. The rpesented data support the conclusions. For these reason, the manuscript can be acceprted for publication, provided that the below listed minor comemnts are solved.
As a final note, a general improvement of the language is needed. In particular (see minor note 1 as example), the use of plural must be preferred to singular form everywhere in the manuscript. Also, avoid within all the manuscript the use of first person (for instance, line 78 “we will propose..”)

→ We appreciate considerate advices of the reviewer 2. We thoroughly revised the manuscript including the points raised by the reviewer 2.

Minor notes:
1) line 39 use “dyes are … agents” instead of “dye is … agent”

→ We revised as reviewer’s comment form “dye is … agent” to “dyes are … agents” in line 39.

2) Line 43 “labeled on” is not the correct term: “bound to”?

→ We revised as reviewer’s comment form “labeled on” to “bound to” in line 43.

3) Line 75-76 “field-75 emission scanning electron microscopy (FE-SEM),” is cited while in the abstract trasmision EM is cited: In experimental both trasmission and scanning: please clarify and make consintent all the sections

→ We appreciate kind advice of the reviewer. We revised manuscript to make more clear.

4) Line 78: “from 0.5 h to 24 h” explain better;

→ The sentence is revised from “from 0.5 h to 24 h” to “on designed time points (0.5, 1, 2, 3, 6, 12 and 24 h)”

5) Line 83 “Ga(NO3)3·xH2O” what about “x”? Please explain the meaning

→ The water contents in gallium nitrate hydroxide is not defined by provider (Sigma-Aldrich Co. LLC.). All reagent in this research were used without any purification.

6) Line 132 The partial dissolution of crystallites during LDH anion exchange reactions was demsotrated by Palin et al., 2019, Understanding the Ion Exchange Process in LDH Nanomaterials by Fast In Situ XRPD and PCA-Assisted Kinetic Analysis, Journal of Nanomaterials, https://doi.org/10.1155/2019/4612493: the work, where the kintic of the processi s also discussed, should be cited and commented here and/or in the discussion

→ According to the reviewer’s comment, we carefully revised manuscript and added recommend reference.

7) Line 149 “insertion” instead of “stabilization”?

→ We revised the word “stabilization” to “stable”.

8) Line 154: in the caption of figure 1, please write the technique used for Ga quantification

→ As reviewer suggested, we revised the caption for Figure 1 with detail.

9) Line 177: plase in the caption of table 1 plese recall the maning on sample names “Ga@LDH-x”. i.e the meaning of the number “x”; in the text recall how these samples were produced, maybe referring to the experimental section. Remove the first decimal digit from crystal sizes and explain that this is an estimation, known all the limitations of Sherrer approach, expecially when applied to layered materials. Please comment the differences with sizes from electron microscopy

→ According to reviewer’s comment, we revise the caption of Table. 1 with detail information of sample name. Also remove the first decimal digit from crystal sizes.

10) Line 180 the title of section 3.3 is misleading. Maybe “Crystalliniy and purity analysis”?

→ We revised the title of section 3.3 from “Crystal structure analysis” to “Crystal structure and crystalline impurities analysis”.

Round 2

Reviewer 1 Report

I satisfied with the responses and corrections done by the authors.